# Resolvases, Dissolvases, and Helicases in Homologous Recombination: Clearing the Road for Chromosome Segregation

**DOI:** 10.3390/genes11010071

**Published:** 2020-01-08

**Authors:** Pedro A. San-Segundo, Andrés Clemente-Blanco

**Affiliations:** Institute of Functional Biology and Genomics (IBFG), University of Salamanca and Consejo Superior de Investigaciones Científicas (CSIC), 37007 Salamanca, Spain

**Keywords:** helicases, dissolvases, resolvases, mitosis, meiosis, DSB repair, homologous recombination

## Abstract

The execution of recombinational pathways during the repair of certain DNA lesions or in the meiotic program is associated to the formation of joint molecules that physically hold chromosomes together. These structures must be disengaged prior to the onset of chromosome segregation. Failure in the resolution of these linkages can lead to chromosome breakage and nondisjunction events that can alter the normal distribution of the genomic material to the progeny. To avoid this situation, cells have developed an arsenal of molecular complexes involving helicases, resolvases, and dissolvases that recognize and eliminate chromosome links. The correct orchestration of these enzymes promotes the timely removal of chromosomal connections ensuring the efficient segregation of the genome during cell division. In this review, we focus on the role of different DNA processing enzymes that collaborate in removing the linkages generated during the activation of the homologous recombination machinery as a consequence of the appearance of DNA breaks during the mitotic and meiotic programs. We will also discuss about the temporal regulation of these factors along the cell cycle, the consequences of their loss of function, and their specific role in the removal of chromosomal links to ensure the accurate segregation of the genomic material during cell division.

## 1. Introduction

Homologous recombination (HR) is central to the maintenance of genome integrity in all eukaryotic organisms. Mirroring this importance, HR is essential to guarantee the efficient restoration of the lost information during the repair of mitotic DNA breaks, to facilitate the recovery of stalled replication forks and to ensure the correct distribution of chromosomes to the gametes during meiosis. In response to a spontaneous mitotic DNA double-strand break (DSB), HR drives the repair of the lesion by ensuring the identification of a homologous sequence, preferentially in the sister chromatid, that will act as a donor for the accurate recovery of an intact DNA molecule after a strand-exchange reaction. On the other hand, after the formation of meiotic DSBs by the Spo11 complex, HR leads to the establishment of physical connections among homologs that are crucial to sustain proper meiotic chromosome segregation and to enhance the genetic diversity of the gametes. In both situations, HR is initiated by the activation of resection, a process that involves bidirectional nucleolytic degradation of the two sides of the DSB, leading to the formation of single-stranded DNA (ssDNA) tails that stimulate homologous pairing and strand invasion with a DNA template (Figure 1). Resection is executed by two sequential events that depend on the activation of different nucleases. The initial processing is carried out by the MRX complex (Mre11-Rad50-Xrs2) that together with Sae2 produces a short 3′ overhang ssDNA [1]. In a subsequent step, two redundant pathways involving the Dna2/Sgs1-Top3-Rmi1 complex and the exonuclease Exo1 catalyze a long-range resection processing, leading to the formation of extended ssDNA tails at both sites of the DNA lesion [2]. Importantly, while both pathways cooperate in parallel during the resection of a mitotic DSB, only Exo1 is required for full resection during the repair of meiotic DSBs. The extensive ssDNA tracks generated during resection are quickly covered by the heterotrimeric RPA complex during both mitotic and meiotic DSB processing. It has been proposed that the main role of RPA is to protect the ssDNA and to remove secondary structures created during the execution of the resection process. Interestingly, the binding of RPA at the vicinity of the DSB also serves as a platform for the recruitment of components of the checkpoint machinery, such as Mec1/Ddc2, to signal the presence of unrepaired DSBs triggering the appropriate DNA damage response [3,4,5,6,7]. In addition, the highly-conserved Rad51/Dmc1 recombinases that promote homology search and strand invasion are also recruited to the RPA-coated ssDNA filament. Rad51 assembles into extended helical filaments on ssDNA to form the nucleoprotein filament, a structure with the capacity to search for a dsDNA donor sequence and promote base pairing. While Rad51 is involved in both mitotic and meiotic recombination, Dmc1 is a specialized player of HR in meiosis. Dmc1 specifically promotes the invasion of a DNA template into a non-sister chromatid to ensure the formation of inter-homolog connections, essential to drive accurate chromosome segregation to the meiotic progeny [8,9]. Interestingly, during meiotic recombination Dmc1 is in charge of the strand-exchange reaction and Rad51 only acts as a Dmc1-accessory factor [10]. It has been proposed that Dmc1 specialization in meiosis has been evolutionarily selected for its capacity to stabilize mismatched base triplets, a feature that allows this recombinase to promote recombination between different parental alleles [11,12].

Independently of the DSB repair program used, the Rad51/Dmc1 nucleofilament has the ability to test homology with an intact DNA donor sequence and promote strand exchange, leading to the formation of the displacement loop (D-loop) (Figure 1). D-loop formation is subjected to dynamic reversibility, a process that has been proposed to contribute to the fidelity of HR [13]. The invading strand of the D-loop is extended by DNA polymerases that copy the lost information at the break site. Once the lost DNA sequence has been copied from the donor strand, the D-loop can be processed throughout two different sub-pathways: Synthesis-Dependent Strand Annealing (SDSA) or Double Strand Break Repair (DSBR). In SDSA, the invading strand of the DNA is displaced and re-annealed to the other broken chromosome. In DSBR the invading strand will be extended in the D-loop while the displaced strand will capture the other DSB end to ultimately form a closed covalent DNA intermediate known as the double Holliday junction (dHJ) that presents two physical connections between the two molecules of DNA involved. Depending on the sub-pathway used for repairing the DSB, different recombinant outcomes can appear: While SDSA generates only non-crossovers (NCOs), both crossovers (COs) and NCOs may arise from DSBR (Figure 1). In some circumstances, an alternative pathway resulting from the nuclease-dependent early cleavage of the D-loop can also generate crossovers [14,15,16,17]. A number of excellent reviews thoroughly covering all the mechanistic details of HR pathways are available [18,19,20,21,22,23]. Controlled disengagement of the joint molecule (JM) intermediates generated by HR must occur prior to chromosome segregation to guarantee the equal distribution of intact genomic information during cell division. Two distinct biochemical activities are capable of dismantling these DNA linkages: helicase-mediated displacement and nucleolytic resolution. In this review, we will mainly focus on the regulation and biological relevance of the helicases Srs2 and Mph1, the dissolvase complex Sgs1-Top3-Rmi1 (STR) and the resolvases Mus81-Mms4, Yen1, Slx1-Slx4, and Mlh1-Mlh3-Exo1. These enzymes are specialized in taking apart all obstacles generated during DSB repair in mitosis and meiosis that could interfere with the accurate segregation of budding yeast chromosomes. The role of these enzymatic complexes in tackling the problems emerging from replication stress has been the topic of other reviews [24,25] and therefore will not be extensively discussed in the present article. Finally, taking into account that the basis of most of the biological processes covered here are conserved from yeast to humans, we will mainly focus on the knowledge obtained in the model system *Saccharomyces cerevisiae*.

## 2. Processing of Recombination Intermediates by Helicases

### 2.1. The Srs2 Helicase-Translocase

In mitotic cells, most DSBs are principally repaired by the SDSA pathway. The use of SDSA minimizes the generation of interhomolog COs that could lead to loss of heterozygosity. Probably, the most well-known factor involved in dismantling D-loop structures is the ATP-dependent helicase/translocase Srs2 of the UvrD family [26]. Srs2 was firstly identified as a suppressor of *rad18Δ* mutants that channeled DNA lesions into a gap-repair pathway. Surprisingly, its absence leads to a hyper-recombination phenotype, showing for the first time the anti-recombinational properties of this protein [27]. Today we know that Srs2 can operate at three levels: by stripping Rad51 from the nucleofilament, by reversing nascent D-loops to reinforce accuracy in HR and, finally, by disrupting extended D-loops enabling SDSA (Figure 1). Therefore, Srs2 possesses both anti- and pro-recombinogenic functions [28,29,30,31]. The anti-recombinogenic activity of Srs2 is achieved by its ability to remove Rad51 from ssDNA [32,33]. Supporting this view, Srs2 contains a highly-processive translocase activity capable of efficiently stripping Rad51 from the nucleofilament and disrupt heteroduplex intermediates [28,29,31]. It has been postulated that the “strippase” activity of Srs2 on Rad51 is vital to remove toxic recombination intermediates generated during the repair process, since deletion of *RAD51* rescues most of the lethality and growth defects associated to *srs2Δ* in combination with other repair mutants [27,34,35,36]. At the biochemical level, it is believed that Srs2 antagonizes Rad51 binding to ssDNA by enhancing the ATPase activity of Rad51, thus activating its turnover from chromatin and favoring its exclusion from the ssDNA nucleofilament [37,38]. In line with these evidences, the Rad55/Rad57 complex promotes Rad51 binding to ssDNA [39] by counteracting the strippase activity of Srs2 [40]. Interestingly, it has recently been shown that Srs2 has also the ability to remove and redistribute RPA, Rad52 and DNA polymerase δ on the ssDNA, suggesting that the capacity of Srs2 to act on recombination factors is not protein specific [30,41]. Finally, it is important to remark that Srs2 also enhances the resolution of recombination intermediates independently from its strippase function. In this regard it has been postulated that Srs2 interacts with Mus81-Mms4 and stimulates its nuclease activity in a helicase-independent manner on a great variety of DNA substrates [42].

At the DNA repair level, it is well known that Srs2 promotes NCO outcomes by biasing DNA repair to the SDSA pathway [43] (Figure 1). Thus, the lack of Srs2 diminishes SDSA events and increases the proportion of COs associated to DSB repair. This effect in restraining CO formation is in part due to its role in counteracting Rad51 binding to ssDNA. However, this is not the only function that Srs2 exerts during DNA repair to avoid unwanted recombination intermediates. It has been established that recruitment of Srs2 to SUMO-modified PCNA during the DNA damage tolerance response constitutes an additional mechanism to avoid excessive recombinational activity [44,45]. Supporting this observation, the Srs2 carboxy-terminal domain contains tandem receptor motifs that are required for the recognition by PCNA-SUMO [46]. Accordingly, the Rad51 binding domain of Srs2, as well as its PCNA- and SUMO-interaction motifs are essential to promote SDSA, thus preventing CO formation [30,47]. Interestingly, phosphorylation of Srs2 by Cdk1 also constitutes a molecular switch that modulates its activity (Figure 2). Cdk1-dependent phosphorylation of Srs2 is required for accurate DSB repair, likely by controlling the disassembly of D-loop structures [48]. However, the lack of Srs2 phosphorylation neither affects Rad51 removal from the nucleofilament nor PCNA interaction, but leads to the accumulation of SUMOylated Srs2. These results suggest that Cdk1-dependent phosphorylation of Srs2 is responsible for counteracting its unscheduled SUMOylation, thus promoting the dismantling activity of Srs2 on D-loops in a helicase-dependent manner [48].

The involvement of Srs2 in the repair of mitotic DNA lesions can also be extended to meiotic DSBs, although much less is known about the functional contribution of Srs2 during meiosis. Indeed, *srs2Δ* mutants have reduced spore viability, delayed meiotic progression and lower levels of CO and NCO formation [49,50]. Recently, it has been demonstrated that the lack of Srs2 precludes normal chromosome segregation during meiosis, generating asci with unseparated and fragmented DNA masses, and nuclei connected by chromatin bridges [51,52]. Importantly, *srs2Δ* cells accumulate Spo11-dependent Rad51 aggregates at the end of meiotic prophase I that colocalize with RPA. This result suggests that the meiotic role of Srs2 in avoiding unwanted recombination intermediates also involves limiting Rad51 accumulation on DNA [51,52]. Nevertheless, the precise nature of the persistent aberrant recombination intermediates in *srs2Δ* meiotic cells is still unknown. It is important to remark that since interhomolog CO recombination is a hallmark of meiosis required for accurate chromosome segregation, multiple mechanisms exist to reinforce CO formation. In this regard, it has been shown that the meiosis-specific recombinase Dmc1 limits Srs2 function by inhibiting its ATP hydrolysis activity, thus preventing its translocation on Dmc1-bound ssDNA filaments and enhancing the formation of desired interhomolog CO recombination events [53]. Highlighting the relevance of Srs2 function in meiosis, a complex network of multiple protein interactors during meiotic prophase I, including factors involved in DNA and RNA metabolism, has been recently discovered [54]. Whether these novel Srs2-interacting proteins are related to the processing of recombinant joint molecules (JMs) remains to be established.

### 2.2. The Mph1 Helicase Contributes to D-Loop Dismantling

Mph1, the homolog of Fanconi anemia protein M (FANCM), was originally identified in budding yeast as a factor that prevents spontaneous mutagenesis [55]. It was soon realized that this function depends on the helicase activity of the protein, whose absence exacerbates the appearance of missense mutations. The first evidence for a role of Mph1 in the processing of JMs came from analysis of the genetic interactions of *mph1Δ* with different HR mutants demonstrating its involvement in the recombinational pathway [56]. Mph1 is an ATP-dependent helicase with 3′ to 5’ polarity [57]. Like other DNA damage-related helicases, the main role of Mph1 in response to a DSB is to act as an anti-recombinase factor. This function was initially suggested when realizing that *MPH1* overexpression restrains HR and delays Rad51 recruitment to DSBs [58]. Supporting this observation, depletion of both Mph1 and Srs2 synergistically sensitizes cells to DNA damage induced by methyl methanesulfonate (MMS) [58]. These findings suggest a role for Mph1 in counteracting HR to avoid undesired recombinational events that could lead to the appearance of gross chromosomal rearrangements. Analysis of the impact of Mph1, Srs2, and Sgs1 on CO-NCO formation in a gap-repair assay has revealed that the mechanism of action of these helicases seems to be different [59]. This study also led to the conclusion that, in addition to D-loops, Sgs1, and Srs2, but not Mph1, also disassemble HJ-containing intermediates [59]. In vitro assays of D-loop reconstitution showed that Mph1 is extremely efficient in displacing these structures [60], a feature that supports a role for the helicase in promoting SDSA in vivo [13,61]. This Mph1-dependent bias to SDSA is fundamental to avoid the processing of recombination intermediates by nucleases that can contribute to the generation of COs (Figure 1). In this regard, Mph1 has been proposed to prevent CO formation by removing substrates for Mus81-Mms4 [61]. It is important to remark that Mph1 helicase activity is also crucial to prevent the accumulation of RNA-DNA hybrid-dependent DNA damage, expanding the role of this helicase to the disentanglement of R-loop structures [62].

Like the Sgs1 helicase (see Section 2.3), Mph1 is also subjected to regulation by the Smc5/6 complex. It has been postulated that Smc5/6 is required to facilitate the resolution of recombination intermediates generated by Mph1 [63]. Interestingly, the role of the Smc5/6 complex in the regulation of Mph1 is distinct to that observed in Sgs1, as *mph1Δ* does not alleviate the accumulation of recombination intermediates generated in cells lacking Sgs1 activity [63]. Smc5/6 also controls the action of Mph1 at RNA-DNA hybrids [62]. In addition to the role of the Smc5/6 complex in controlling Mph1 activity, Mte1 can also regulate the function of the helicase. It has been demonstrated that Mte1 binds directly to D-loops [64,65] through previously bound Mph1 to collaborate in the disappearance of Rad52 foci during DSB repair [66].

Much less is known about the possible role of Mph1 during meiosis in *S. cerevisiae*. A recent comprehensive proteomics work aimed to discover the interaction profile of various DNA-processing enzymes involved in HR has revealed that the Mph1 protein interaction landscape is significantly reduced in meiotic versus mitotic cells [54]. Nevertheless, this study provides evidence for the existence of protein complexes including Mph1 and other genome integrity factors (Mhf1, Mhf2, and Mte1), as well as transcription factors (Fkh1, Fkh2) during meiotic prophase I. Investigation of the functional relevance of these interactions during meiotic DSB repair may unravel unknown aspects of this process. Indeed, and consistent with their relationship in mitotic cells, Mph1 and Mte1 also affect the DSB strand exchange step antagonistically during meiotic recombination. These functions are particularly critical when nucleus-wide DSB abundance is low, e.g., during early meiosis (V. Börner, personal communication; [67]).

### 2.3. The Dissolvase Role of the Multifaceted STR Complex

The dissolution of the HJ is one of the most important and delicate mechanisms involved in the disassembly of recombinant structures into NCO repair outcomes. Probably, one of the most studied factors involved in this process is Sgs1, a helicase of the RecQ family in *S. cerevisiae.* Sgs1 represents the homolog of human BLM, defects in which cause the cancer-prone Bloom´s Syndrome. The multiple phenotypes associated to the lack of Sgs1 activity mirror its importance in various cellular events, such as DNA replication, checkpoint response and recombination [68]. The *sgs1* mutant was initially discovered as a slow growth suppressor of *top3* mutants, leading to the suggestion that Sgs1 produces the substrates for Top3 [69]. Sgs1 is a 3′ to 5′ helicase that associates to Top3 and Rmi1 to form the STR complex. While Sgs1 contains a potent DNA helicase activity, Top3-Rmi1 form a Type IA DNA topoisomerase that generates a transient DNA nick that allows the transfer of a second unbroken DNA strand through the nick. It has been postulated that the role of Rmi1 in this process is to stabilize the open conformation of Top3 to bias decatenation over relaxation [70,71]. The most accepted model that interconnects both Sgs1 and Top3-Rmi1 requirements for the resolution of recombinant intermediates is that Sgs1 would use its unwinding activity to remodel DNA substrates into accessible conformations for the Top3-Rmi1 dimer.

The STR complex is involved in multiple steps of the repair process, having both pro- and anti-recombinogenic functions. Sgs1 promotes HR by enhancing long-range DSB end resection in an Exo1-independent Dna2-dependent pathway [2,72] (Figure 1). It has been established that both Sgs1 and Dna2 physically interact, and that the helicase activity of Sgs1 unwinds the DNA and feeds ssDNA to Dna2 for degradation [73]. Interestingly, while Top3 also participates in this process, its catalytic activity is not required [74]. Notably, Sgs1 is dispensable for full-length meiotic DSB resection, which mainly relies on Exo1 and Tel1 activities [75]. On the other hand, the STR complex, via Top3 decatenation activity, is capable of dismantling D-loops in vivo, favoring the formation of NCOs by the SDSA repair pathway both in the meiotic [76,77] and mitotic [13] programs (Figure 1). Although in vitro assays using artificial D-loops suggested that Top3 preferentially dismantles protein-free D-loops [78,79], more recent single-molecule imaging analysis has revealed that Sgs1 possesses the ability to disrupt Rad51-bound ssDNA nucleofilaments [80]. Like Srs2 [50,53], Sgs1 cannot act on Dmc1-coated ssDNA filaments, but unlike Srs2, the mechanism employed by the Sgs1 helicase to remove Rad51 is independent of the Rad51 ATPase cycle [80]. Top3-Rmi1 can also dissolve Rad51-mediated D-loops via a topoisomerase-dependent mechanism without the participation of the Sgs1 helicase [78]. In a late step of the HR pathway, the STR complex participates in the dissolution of mitotic dHJs resulting from HO-induced DSBs to promote NCO products [43].

In meiosis, the STR complex is essential to ensure the normal pattern of NCO and CO meiotic recombination outcomes [76,77,81,82,83,84]. On one hand, due to the D-loop reversal activity, STR prevents the formation of promiscuous aberrant recombination intermediates involving multichromatid joint molecules (mcJMs). In addition, the STR complex has also the capability to dissolve dHJs that have escaped the stabilization by ZMM proteins. Notably, while Top3-Rmi1 participates in all Sgs1 functions during meiotic recombination, Top3-Rmi1 also displays a distinct Sgs1-independent late role to resolve complex recombination intermediates that would otherwise prevent chromosome segregation [85,86] (Figure 1). In fact, recent work using return-to-growth (RTG) experiments has shown that the absence of Rmi1, but not that of Sgs1, prevents complete JM resolution leading to the activation of the Rad9-dependent DNA damage response [87]. Consistent with this notion, only a small fraction of the Top3-Rmi1 complex present in prophase I cells is stably associated with Sgs1 [54].

In accordance to the multiple roles of the STR complex along the numerous pathways contributing to the maintenance of genome integrity, its activity is tightly regulated during DNA repair. Recently, the Smc5/6 complex has emerged as a new STR regulator during the repair of DNA lesions. The first evidence suggesting a crosstalk between the STR and Smc5/6 complexes came from the observation that *smc5/6* mutants share most of the phenotypes associated to the lack of Sgs1 [88,89,90,91]. Now we know that the Smc5/6 complex regulates Sgs1 activity in a dual step mechanism. In a first stage, the Mms21 SUMO-ligase component of the Smc5/6 complex SUMOylates Smc5/6 subunits at DNA damage sites. In this situation the STR complex is recruited to SUMOylated Smc5/6 via two SIM motifs present in Sgs1. Once recruited, STR is activated by Mms21-dependent SUMOylation of its subunits [92,93]. Interestingly, the Smc5/6-Mms21 complex is necessary for antagonizing recombination intermediates generated in meiosis by destabilizing early intermediates and resolving JMs, extending the role of this mechanism to the meiotic pathway [94,95,96]. In addition to SUMO-dependent regulation, it has been shown that Sgs1 might also be subjected to phosphorylation-dependent regulation during the prophase to metaphase I transition in meiotic cells, coincident with the induction of Cdc5 [54]. On the other hand, it has been established that Sgs1 phosphorylation by Mec1 promotes its binding to the FHA1 domain of Rad53, enhancing its phosphorylation in response to replicative stress [97]. Moreover, elevated levels of Sgs1 phosphorylation are detected in damaged cells lacking the PP4 phosphatase, suggesting that its activity is tightly regulated by phosphatases and kinases working in tandem during the DNA damage response [98].

## 3. Nucleolytic Resolution of Holliday Junctions

### 3.1. The Mus81/Mms4 Complex: A Crucial Actor in Joint Molecule Resolution

Mus81 was originally identified in a two-hybrid screen using Rad54 as a bait. Besides this physical interaction, the sensitivity of the *mus81Δ* mutant to several genotoxic stresses suggested for the first time the involvement of Mus81 in recombinational repair [99]. The first evidence for the role of the Mus81-Mms4 complex in the resolution of recombinant DNA structures came for the observation that the function of this heterodimer was essential for viability of cells lacking the activity of the STR complex [100]. In fact, both Mus81-Mms4 and STR act over the same 3′ flap recombination intermediates [101]. The Mus81-Mms4 heterodimer is an evolutionarily conserved structure-selective endonuclease (SSE) related to the Rad1-Rad10 (XPF/ERCC1) family. While Mus81 contains the catalytic activity of the complex, most of the regulatory motifs of the heterodimer are found in Mms4 [102]. Still, both subunits are essential for substrate recognition and nuclease activity of the complex [100]. The enzymatic characterization of the purified complex in vitro demonstrated that Mus81-Mms4 is catalytically active over a great variety of substrates, including fork-like replication structures, nicked Holliday junctions and D-loops [14,103]. One of the first in vivo observations that involved the Mms81-Mms4 complex in the resolution of recombination intermediates came from RTG experiments, where *ndt80Δ* meiotic cells that accumulate unresolved JMs are returned to the mitotic cell cycle by a nutritional shift. These experiments confirmed that in mitosis most JMs are dissolved by Sgs1 to produce NCO outcomes, but those recombination intermediates that escape the activity of the helicase are processed later on by the Mus81-Mms4 complex to produce both CO and NCO products [104,105] (Figure 1). The action of Mus81-Mms4 as a fail-safe mechanism to resolve DNA intermediates that cannot be processed by other enzymatic complexes has been also reported in the response to replication stress [105]. Accordingly, the lack of Mus81-Mms4 activity leads to the formation of spontaneous unresolved sister chromatid anaphase bridges that depend on the homologous recombination pathway [106].

Structure-selective endonucleases can be considered as double-edged swords: they are essential to process chromosome linkages in different situations allowing accurate segregation of intact chromosomes, but they must be tightly regulated to prevent their premature action with lethal consequences. Mus81-Mms4 regulation during the cell cycle is achieved at various levels. The sequential activation of Sgs1 and Mus81-Mms4 is controlled by the phosphorylation status of their components. Thus, Mus81-Mms4 is hyperactivated by Cdc5, CDK and DDK-mediated phosphorylation of the Mms4 regulatory subunit in G2/M, a feature that stimulates the complex as cells enter anaphase [103,107,108,109,110] (Figure 2). Mirroring the importance of this temporal control, a premature activation of Mus81-Mms4 attained by expressing a phosphomimetic version of Mms4 induces CO-associated chromosome translocations and precocious processing of recombinant intermediates [111]. Confirming these results, uncontrolled Cdc5 production increases CO frequency and the potential for loss of heterozygosity due to the premature activation of Mus81-Mms4 [109]. These results suggest that post-replicative stimulation of the Mus81-Mms4 endonuclease avoids its full activation during S-phase, thus preventing potential cleavages on DNA structures formed during replication. Interestingly, the S-phase steady state activity of Mus81-Mms4 is enough to suppress template switching between homologous sequences by restricting error-prone DNA synthesis during HR at broken replication forks [112]. This observation suggests that the residual S-phase activity of Mus81-Mms4 is needed for the resolution of unscheduled recombination intermediates generated during replication. Supporting this hypothesis, it has been shown that, at least in *S. pombe* and mammalian cells, the spontaneous cleavage of stalled forks by Mus81 is allowed only when the replisome is not stabilized by the replication checkpoint and is essential for recovery of stalled replication forks [113,114,115].

Another regulatory layer on the control of Mus81-Mms4 is imposed by its interaction with other recombination factors. It has been established that Rad54 strongly stimulates Mus81-Mms4 nuclease activity in an ATP binding/hydrolysis-independent manner by targeting the complex to its DNA substrates [116]. Interestingly, Rad54 is not the only factor involved in the regulation of the Mus81-Mms4 complex. It has been described that Srs2 and Mus81-Mms4 physically interact and act in a coordinated way to process recombination intermediates. Srs2 promotes Mus81-Mms4 function both by directly stimulating the nuclease activity in a helicase-independent manner and by removing Rad51 from DNA allowing Mus81-Mms4 access to the substrates. On the other hand, Mus81-Mms4 restrains Srs2 helicase activity [42]. Another positive regulator of the Mus81-Mms4 complex is the RENi family member Esc2. It has been demonstrated that Esc2 binds to Mus81-Mms4 via its SUMO-like domains, stimulates its enzymatic activity in vitro and participates in the resolution of DNA junctions generated during the replication of damaged DNA [117].

Finally, it has been proposed that the activity of Mus81-Mms4 can be also regulated by controlling its subcellular localization. Following DNA damage, Mus81-Mms4 relocalizes to a class of subnuclear stress foci defined by Cmr1/WDR76. In the presence of recombination intermediates, Mus81-Mms4 foci persist until the nuclease is activated by Mms4 phosphorylation. Interestingly, other SSEs, such as Rad1-Rad10 and Slx1-Slx4, also colocalize with Mus81-Mms4 in these subnuclear foci suggesting that relocalization of SSEs could constitute another layer of regulation aimed not only to promote the efficient resolution of DNA intermediates, but also to prevent their untimely function [118].

Regarding the meiotic program, it was soon discovered that the Mus81-Mms4 complex is also involved in meiotic recombination [100]. The early characterization of cells lacking Mms4 shed light into the meiotic role of Mus81-Mms4 by implicating the complex after strand invasion and pinpointing its requirement for processing certain recombination intermediates generated during meiosis [119]. Later, it was discovered that the Mus81-Mms4 complex is specifically required for the production of non-interfering class II COs [120,121]. This function in the removal of meiotic recombination products overlaps with the role of the Sgs1 helicase, since double mutants lacking both Sgs1 and Mus81 show a strong accumulation of meiotic recombination intermediates [81]. Although both STR and Mus81-Mms4 collaborate in the elimination of aberrant JMs, they operate in temporal and mechanistically distinct pathways [81,122]. Unprotected D-loop intermediates are disassembled by STR to yield NCOs by SDSA. Strand-invasion events captured by the ZMM complex originate dHJs that are resolved to interfering type I COs by the MutLγ-ExoI resolvase (see Section 3.4). In turn, ZMM-independent JMs that escape STR dissolution are resolved by SSEs, primarily Mus81-Mms4, to form both NCOs and type II COs [76,77] (Figure 1). Ndt80-dependent expression of Cdc5 promotes resolution of dHJs prior to chromosome segregation [123]. Importantly, Cdc5 phosphorylates and activates Mus81-Mms4 to promote type II COs [108] (Figure 3). However, how Cdc5 drives the resolution of MutLγ-Exo1-dependent COs remains unknown.

### 3.2. Yen1: The Last Chance to Resolve DNA Linkages

The evolutionarily conserved Yen1 protein was first discovered in rice and flies as a Rad2- family nuclease with a broad substrate range [124,125]. The budding yeast Yen1 protein was identified by screening a yeast gene fusion library for nucleases capable of resolving HJ structures [126]. It was described that this SSE is able to resolve HJs in vitro by introducing symmetrical cuts across the junction point to generate nicked duplex products that can be ligated [126]. Genetic analyses have shown that while a *yen1Δ* mutant maintains cell viability in response to a great variety of DNA-damaging agents that affect replication fork progression, a *yen1Δ mus81Δ* double mutant is extremely sensitive to these genotoxic stresses [127,128,129]. Importantly, this synthetic lethality is bypassed by deleting *RAD52*, indicating that the accumulation of toxic recombination intermediates is the cause of the cell viability loss observed in the absence of Yen1 and Mus81 upon DNA damage [127]. These data strongly support the concept that Yen1 operates together with Mus81-Mms4 for the correct resolution of recombinant intermediates generated during HR (Figure 1). This hypothesis was indeed confirmed in an assay to detect unselected products of mitotic recombination that demonstrated that both Mus81-Mms4 and Yen1 collaborate in the resolution of recombination intermediates generated after induction of a DSB [128]. Interestingly, elimination of both Mus81 and Yen1 bias the DNA repair process into a Pol32-dependent break-induced replication (BIR) pathway, resulting in a significant increase in the frequency of spontaneous loss of heterozygosity events and chromosome missegregation [128,130]. Paradoxically, in an analysis to detect DSB-induced chromosomal translocations, it was observed that both Mus81 and Yen1 enhance BIR both by allowing the establishment of a replication fork and by promoting template switching to produce non-reciprocal translocations [131].

Even though it seems clear that Yen1 exhibits a redundant role with Mus81-Mms4 during the resolution of recombinant intermediates, it has been demonstrated that these nucleases contain an exquisite selectivity for various DNA JMs [130]. Therefore, it is tempting to speculate that each nuclease might be involved in the resolution of specific recombinant substrates during DNA repair. Supporting this notion, it has been postulated that the DNA recombinants generated during the treatment with alkylating agents in the absence of a proficient STR complex are processed by Mus81-Mms4, but not by Yen1 [132]. Moreover, Yen1 and Mus81-Mms4 work independently in two different resolution pathways for the repair of replication-born DSBs by sister chromatid recombination [133]. These results suggest that both Mus81-Mms4 and Yen1 might account for the resolution of different recombinant structures.

At the regulation level, Yen1 activation/deactivation has been the focus of increasing interest during the last years. The first evidence of a possible cell cycle regulation of its activity was obtained when Yen1 appeared in a systematic screen headed to identify CDK-regulated nucleo-cytoplasmic shuttling proteins by using a prediction system for nuclear localization signals (NLSs) [134]. Later on, it was demonstrated that Yen1 is indeed phosphorylated by CDK on its NLS during S-phase to promote its nuclear exclusion and, consequently, its inhibition. In anaphase, activation of the Cdc14 phosphatase drives Yen1 dephosphorylation allowing its translocation to the nucleus and reactivation of its enzymatic activity [135,136,137]. The importance of this tight control of Yen1 activity was manifested in experiments employing phospho-deficient variants of the protein. Constitutively active Yen1 phospho-deficient mutants (*YEN1^ON^*) develop an increase in CO formation and loss of heterozygosity in cells lacking Mus81-Mms4 activity [135,136]. This spatial and temporal regulation of Yen1 prevents its premature activation and restricts Yen1 function to the end of mitosis (Figure 2). This favors the exclusive disentanglement of those chromosomal linkages left unresolved by repair mechanisms acting earlier in the cell cycle. Accordingly, activation of Yen1 in late anaphase is sufficient to remove anaphase DNA bridges formed by unresolved sister chromatid links facilitating chromosome segregation [106]. This temporal regulation ensures the resolution of unprocessed recombination intermediates only when other NCO pathways have not been successfully accomplished to prevent far worse genome instability due to chromosome missegregation events. It is important to remark that phosphorylation of Yen1 is not the only posttranscriptional modification that modulates its resolvase activity. It has been demonstrated that in response to DNA damage Yen1 becomes SUMOylated and ubiquitinated by Siz1/Siz2 SUMO ligases and the Slx5-Slx8 ubiquitin ligase complex, respectively. Lack of Slx5-Slx8 maintains Yen1 in a constant SUMOylated state, a situation that stabilizes the protein and relocates it to nuclear foci [138]. In agreement with this SUMO-dependent hyper-activated state of the nuclease, a mutation in the ubiquitin acceptor site (K714) produces an increase in CO formation in response to DNA damage. These results fit with the idea that SUMOylation of Yen1 in response to DNA damage enhances its binding to chromatin, and that Slx5-Slx8-dependent ubiquitination of the nuclease targets it for destruction before replication starts. Therefore, the balance in the SUMO and ubiquitin state of Yen1 constitutes another layer of regulation of the nuclease in response to genotoxic stress.

Various studies have also demonstrated the involvement of Yen1 in the meiotic cycle. Although the *yen1Δ* single mutant does not show any defect in sporulation, the *yen1Δ mms4Δ* double mutant exhibits a drastic reduction in sporulation levels [139] suggesting that different SSEs may have redundant roles during the resolution of meiotic recombination intermediates. In fact, Yen1 collaborates with Mus81-Mms4 and Slx1-Slx4 in the resolution of meiotic JMs that have not been stabilized by ZMM to produce NCOs as well as type-II COs that do not display spatial interference [76,77]. It has been proposed that Yen1 acts as an alternative for Mus81-Mms4 to eliminate persistent JMs [108] (Figure 1). Like in mitotic cells, Yen1 is also the subject of strict spatiotemporal control during meiosis. CDK-dependent phosphorylation of Yen1 inhibits it activity and nuclear localization until the onset of anaphase II to ensure the disentangling of persistent recombination intermediates before the completion of meiosis [108,140] (Figure 3). A constitutively active version of Yen1 refractory to CDK phosphorylation (Yen1^ON^) is prematurely recruited to meiotic chromosomes during prophase I leading to the formation of COs independently of MutLγ, Mus81-Mms4, and STR functions. However, this unrestrained Yen1 activity results in aberrant CO spatial patterning and poor viability of meiotic products [140] highlighting the importance of an strict control of the SSEs involved in the resolution of meiotic recombination intermediates to ensure proper chromosome segregation and preventing the formation of aneuploid gametes.

### 3.3. Slx1-Slx4: A Scaffold for Other Structure-Selective Nucleases

Slx1 is the founding member of a family of proteins that includes *E. coli* YhbQ and *B. subtilis* YazA. This family is characterized by the presence of a UvrC-intron-endonuclease domain (URI) and a C-terminal PHD-type zinc-finger domain [141]. While the nuclease activity of Slx1 by itself is rather weak, its interaction with Slx4 drastically stimulates the activity by 500-fold [130,142]. Slx4 has a SAP domain, a feature that has been reported to mediate DNA-protein interaction, indicating that Slx4 may target the Slx1-Slx4 complex to DNA substrates. At the structural level, it has been shown that Slx1 can form a stable homodimer that blocks its active site. However, Slx1-Slx4 interaction is mutually exclusive with Slx1 homodimerization, suggesting that Slx4 enhances the nuclease activity of Slx1 by just interfering with its homodimerization [143].

The identification of Slx1-Slx4 as a possible complex involved in the resolution of recombination intermediates was obtained in a synthetic-lethal screen to identify proteins that function in the absence of Sgs1 [144]. However, the synthetic lethality of *slx1Δ sgs1Δ* or *slx4Δ sgs1Δ* mutants is not suppressed by the absence of Rad52, indicating that this lethality is not caused by recombination intermediates [101,145]. This observation suggests that either Sgs1 may have several roles or that homologous recombination is needed downstream of the Slx1-Slx4/STR overlapping function [142]. It was soon realized that Slx1 and Slx4 constitute a heteromeric SSE involved in the resolution of branched DNA substrates, particularly simple-Y, 5′-flap or replication fork structures [130,142]. However, its catalytic activity over different recombinant DNA structures might differ from other nucleases. In this line it has been reported that while Slx1-Slx4 can cleave Holliday Junctions in vitro, it is unlikely that this complex could exerts this process in vivo since it cannot cleave HJs symmetrically to form ligatable nicked DNA [142,146]. Independently of the type of DNA substrate catalyzed, the main role of the Slx1-Slx4 heterodimer is to collaborate with Mus81-Mms4 and Rad1-Rad10 complexes to disentangle unresolved JMs generated during DNA repair (Figure 1). Consequently, these complexes colocalize at Cmr1 stress foci in response to DNA damage [118] and interact in genome-wide two-hybrid analysis [147]. Moreover, a mutant defective in Rad1-Rad10-Slx4 fails to repair a DSB generated at the mating type locus, presents an extended cell cycle delay in response to DNA damage and shows decreased viability during mating-type switching [148]. With all this information, it is tempting to speculate that Slx4 might affect the activity or substrate specificity of Rad1-Rad10 to resolve recombination structures generated during DNA repair. In the context of ectopic recombination, Rad1-Rad10 can clip the captured D-loop generating substrates for Mus81-Mms4 cleavage. In the absence of Mus81, JMs containing single HJs are accumulated. These HJs cannot be dissolved by STR and rely on Yen1 for their resolution. Consistent with this model, COs are not recovered from *rad1Δ mus81Δ yen1Δ* triple mutants during recombination between dispersed repeats [61,149].

It is noteworthy to mention that Slx4 can act independently of the Slx1-Slx4 complex. In this regard it has been proposed that Slx4 can form a complex with the BRCA1 C-terminal-domain protein Rtt107 and enhances its phosphorylation, suggesting that both proteins might form part of an active complex involved in the recovery from DNA damage [150]. Accordingly, most of the DNA repair defects observed in *slx4Δ* cells are recapitulated in a strain lacking Rtt107 [150]. Moreover, Slx4 and Rtt107 bind near DSBs to prevent Rad9 recruitment to damage sites, thus limiting checkpoint activation and enhancing resection [151]. Therefore, it seems that one of the Slx1-independent roles of Slx4 involves the attenuation of the DNA damage checkpoint activity in order to elicit a proficient activation of resection during the initial steps of the repair process. Another example of an Slx1-independent role of Slx4 is its function in promoting MMS resistance. It has been shown that cells lacking Slx4 have difficulties in fulfilling DNA synthesis during recovery from replisome stalling induced by MMS in an Slx1-independent manner, suggesting that this factor might have a role in facilitating bypass pathways at stalled forks [152]. Accordingly, Slx4, but not Slx1, is required for the repair of specific replication-born DSBs by sister chromatid recombination, further demonstrating that Slx4 works independently of Slx1 in the repair of DNA lesions that occur during replication [133].

As other nucleases, the Slx1-Slx4 complex is also subjected to phospho-regulation. Slx4 becomes phosphorylated in response to several genotoxic stresses in a Mec1/Tel1-dependent manner [153]. However, a phospho-deficient Slx4 version is fully able to rescue the MMS-hypersensitivity observed in *slx4Δ* cells, indicating that this phosphorylation switch is not controlling all aspects of Slx1-Slx4 functions. On the other hand, it has been demonstrated that Mec1/Tel1-dependent phosphorylation of Slx4 is important to correctly exert single strand annealing, suggesting that some of the roles attributed to this nuclease complex in the DNA damage response are indeed regulated by the phosphorylation state of the complex [152]. Consistent with this notion, Slx4 and Dpb11 form a scaffold complex regulated by phosphorylation that acts in different stages of the cell cycle [154]. During S-phase, the Slx4-Dpb11 scaffold associates with Slx1 and Rtt107 to down-regulate checkpoint signaling and also participates in early stages of template switching to facilitate progression of stalled replication forks. Later on, during M-phase, the Mus81-Mms4 SSE also joins the scaffold complex to promote resolution of JMs [154,155,156] (Figure 2).

Little is known about the function of the Slx1-Slx4 complex in the resolution of recombination intermediates generated in meiosis. It has been proposed that the complex cooperates with Mus81-Mms4 and Yen1 in the alternative pathway that resolves ZMM-independent JMs that have also escaped Sgs1-dependent dissolution, although the contribution of Slx1-Slx4 appears to be minor [76,77] (Figure 1). Like in mitotic cells, Rtt107 and Dbp11 have been identified in Slx1-Slx4 complex purifications from prophase I and metaphase I meiotic cells, suggesting that the physical interactions among Rtt107, Dbp11, and Slx1-Slx4 may have functional implications also during meiosis [54,155,157]. In addition, Slx4 appears to have also an Slx1-independent function in meiotic DSB formation and CO distribution especially in the proximity of centromeres [158]. Interestingly, the Slx4 protein displays an Spo11-dependent mobility shift during early stages of meiosis suggesting that its activity may be also regulated by phosphorylation [54,158].

### 3.4. Mismatch Repair-Independent Role for Mlh1-Mlh3/Exo1 in the Resolution of Recombination Intermediates

The Mlh1 protein of *S. cerevisiae* was initially identified as the *E. coli* mutL homolog involved in the mismatch repair pathway. Disruption of *MLH1* results in elevated spontaneous mutation rates in experiments measuring forward mutation to canavanine resistance and reversion of the *hom3-10* allele [159]. Similar results were obtained when analyzing budding yeast cells lacking Mlh3, indicating that this factor is also required to execute a proficient mismatch repair mechanism [160]. It was soon realized that, in addition to the mismatch repair role, these factors have a second function during meiotic recombination because *mlh1Δ* diploid cells display increased spore lethality and reduced meiotic crossing-over [159,161]. Since both Mlh1 and Mlh3 interact in two-hybrid assays and in co-immunoprecipitation experiments, it was proposed that both proteins form part of a same complex, and the term MutLγ was coined [160,162,163]. Notably, the meiotic crossing over defects of *mlh1Δ* and *mlh3Δ* mutants are similar to those of the *msh4Δ* and *msh5Δ* mutants lacking the meiosis-specific MutSγ complex, resulting in meiosis I homolog nondisjunction and reduced spore viability [161,163,164,165]. MutLγ is considered to be the complex that catalyzes the resolution of ZMM-protected meiotic dHJs into type-I COs [77,166,167,168] (Figure 1). Interestingly, the introduction of a point mutation (D523N) within the conserved putative endonuclease domain motif of Mlh3 confers the same defects as the *mlh3Δ* deletion in meiotic spore viability and CO formation, suggesting that Mlh3 is the catalytic enzyme of the complex [169]. Moreover, Mlh3 also contains a conserved ATP-binding domain, whose mutation affects meiotic CO levels [170]. This suggests that ATP hydrolysis by the MutLγ complex is necessary to execute its function during the resolution of meiotic intermediates. Accordingly, atomic force microscopy has revealed that purified recombinant MutLγ undergoes ATP-driven conformational changes [171].

As other resolvases, MutLγ does not operate alone in the resolution on meiotic recombination intermediates. In fact, MutLγ cooperates with the MutSγ complex formed by Msh4-Msh5 (ZMM components), to promote interference-dependent COs [121]. Furthermore, a direct connection between MutLγ and the exonuclease Exo1 has been proven. This relationship is based on the observation that a *mms4Δ slx4Δ yen1Δ* triple mutant (lacking three SSEs involved in resolving JMs during meiosis) is able to achieve resolution of recombinant intermediates. This capacity depends on Sgs1 in combination with MutLγ and the XPG-family nuclease Exo1 [77]. The function of Exo1 in the removal of meiotic JMs does not depend on its nuclease activity, but requires its interaction with MutLγ, proving a nuclease-independent role of Exo1 in the resolution of dHJs [172]. Taking into account that Exo1 is also required for the resection of Spo11-dependent DSBs, it is intuitive to think that this nuclease might be subjected to a temporal and biochemical control along the different stages of the meiotic pathway. Supporting this hypothesis Mlh1 interacts with Exo1 in meiotic metaphase I cells, but not during prophase I [54]. Recently, several interactors of the MutLγ-Exo1 complex (i.e., Chd1, Rtk1 and Caf120) required for efficient formation of COs have been identified [54]. In particular, this study shows that Chd1 functions at the prophase to metaphase I transition to promote MutLγ-Exo1-dependent COs (Figure 3). Since Chd1 is an ATP-dependent chromatin remodeler, it has been proposed that its motor action may displace nucleosomes to facilitate MutLγ oligomerization on the DNA substrate [166], promoting its nuclease activity for processing CO-designated recombination intermediates [54]. In sum, an exquisite spatiotemporal orchestration of MutLγ-Exo1 along the different stages of meiosis is central to attain the proper landscape of recombination outcomes that ensures the accurate segregation of intact chromosomes to the gametes. Besides the mismatch repair function, a role for MutLγ in the recombinational repair of mitotic DSBs has not been reported so far.

## 4. Concluding Remarks

The processing of recombination intermediates by helicases, dissolvases and resolvases is a fundamental feature of DNA repair. The precise orchestration of these enzymes along the different phases of the mitotic and meiotic cell division ensures the correct segregation of the genome, avoiding the aberrant distribution of the genomic material between different cellular compartments. Our understanding of how different enzymatic complexes are regulated in order to disentangle joint DNA molecules has dramatically grown in the last years. This has been possible not only by the implementation of new genomic and molecular biological approaches, but also by generating new technical strategies to understand the intricate regulation of these enzymes along the cell cycle. However, we are still missing many aspects of the detailed molecular mechanisms that regulate their function. During the last years, a great effort has been made to understand how different enzymatic complexes that deal with recombination intermediates are regulated in time and space. It seems clear that posttranscriptional modification events, such as phosphorylation or SUMOylation of certain subunits, account for one of the more important switches that control their activity along the different steps of DNA repair. Moreover, these modifications also determine the specific activation of each DNA-processing complex depending on the type of DNA lesion produced. In the last years, it has become evident that the spatial regulation of these enzymes is indeed another layer of control that ensures the accurate metabolism of recombination intermediates. Besides, the plasticity in the use of a defined enzymatic complex ensures the correct processing of the multiple and diverse recombinant structures that are generated during the different repair pathways available for the cell. Thus, comprehending the principles behind the regulation of each DNA-remodeling complex required to handle recombination intermediates is fundamental to obtain a complete picture of the different mechanisms that contribute to the maintenance of genomic integrity and the accuracy in the transmission of genetic information to the progeny. As detailed in this review, budding yeast constitutes an excellent model to decipher the intricate molecular mechanisms dealing with chromosome linkages. Indeed, impaired activity of some of these DNA-processing complexes in humans is related to pathogenic outcomes, such as cancer predisposition or reproductive problems. For instance, mutations in *SLX4* have been found in patients with Fanconi anemia clinical features [173,174]. Alterations in BLM and WRN, the human homologs of Sgs1 result in Bloom and Werner syndromes, respectively, characterized by cancer predisposition and developmental defects [175,176,177]. Genetic polymorphisms in MLH1 have also been associated with male infertility [178]. In addition, some of these DNA-remodeling enzymes, such as for example MUS81, are also emerging as potential targets in chemotherapeutic intervention [179,180,181,182]. Thus, advances in yeast studies have important implications in understanding the basis of human disorders linked to genome instability.

## Figures and Tables

**Figure 1 genes-11-00071-f001:**
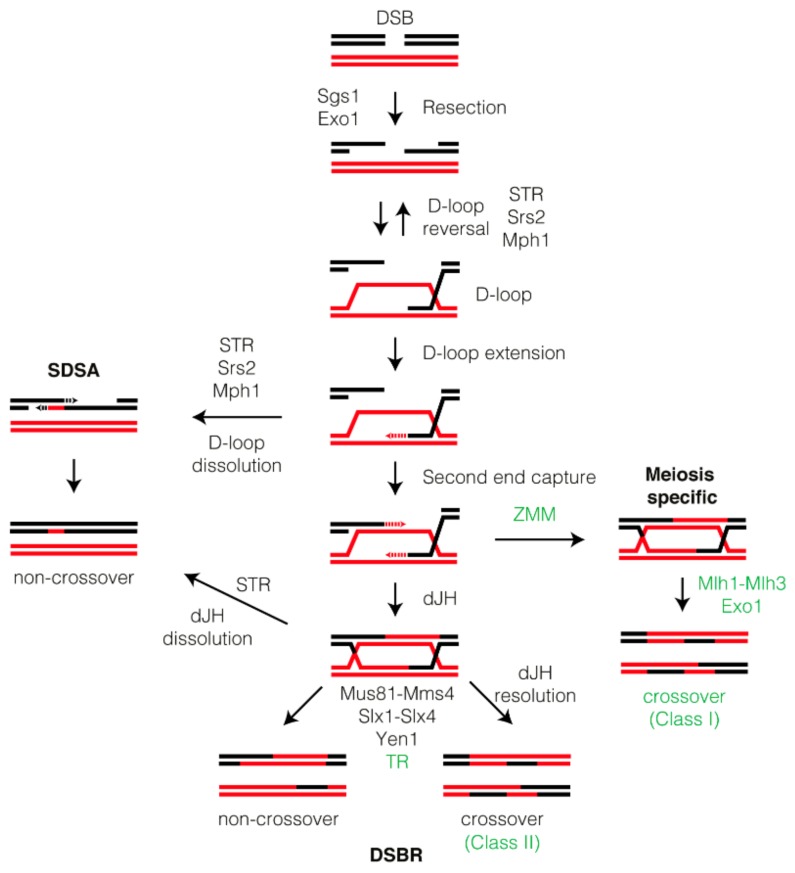
Schematic representation of the role of different helicases, dissolvases and resolvases in the repair of mitotic/meiotic DNA double-strand breaks (DSBs). The diagram represents some of the more relevant DNA structures generated in response to a DSB during the execution of homologous recombination (HR) and the final outcomes produced after repair. Black lettering represents overlapping roles for a specific complex in both the mitotic and meiotic processing of the DSB. Green lettering represents meiosis-specific features. SDSA: Synthesis-Dependent Strand Annealing, DSBR: Double Strand Break Repair.

**Figure 2 genes-11-00071-f002:**
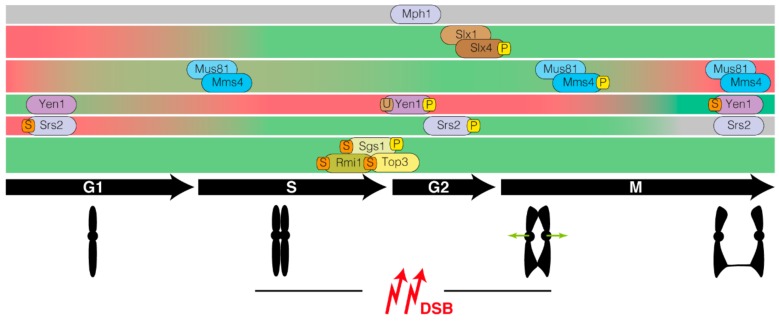
Temporal regulation of DNA-processing complexes involved in the removal of recombinant structures in the mitotic cycle. The diagraph represents the temporal activation (green bars) and inactivation (red bars) of each complex along the cell cycle. Regulation of distinctive enzymatic activities by phosphorylation (P), SUMOylation (S) and ubiquitylation (U) is depicted. Grey bar indicates that the mechanistic details of the temporal regulation are still unknown.

**Figure 3 genes-11-00071-f003:**
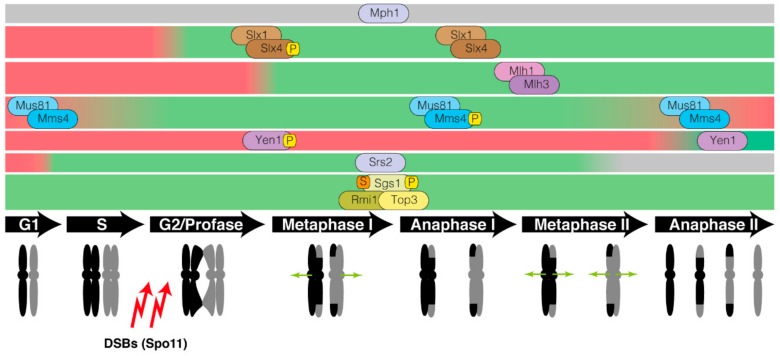
Schematic representation of the temporal regulation of enzymatic complexes involved in the processing of recombinant structures in the meiotic program. The diagraph represents the temporal activation (green bars) and inactivation (red bars) of each complex during meiosis. Different post-translational modifications, such as phosphorylation (P), SUMOylation (S) and ubiquitylation (U) are portrayed. Green and red bars represent active and inactive states, respectively. Grey bar denotes lack of information related to the temporal regulation of the enzymatic complex.

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
