# Peer review of "Resolvases, Dissolvases, and Helicases in Homologous Recombination: Clearing the Road for Chromosome Segregation"

_genes, 2020, doi:10.3390/genes11010071_

Round 1
Reviewer 1 Report
In the present manuscript, the authors review the current state of the art in the field of homologous recombination players involved in the clearing of DNA intermediates that prevent segregation in both mitosis and meiosis.
The paper is well structured and mostly covers the relevant literature. Nonetheless, there are some conceptual major points I want to point out to the authors, and the manuscript requires a sound proof-read for grammar and vocabulary mistakes in english. I will try to help out the authors with a detailed list (not extensive) of such these issues.
Major conceptual points:
1- In the title and throughout the paper, authors talk about "dissolvases" to refer to the helices Sgs1(STR), Mph1 and Srs2. I think that the term should be restricted to the Sgs1 mediated dissolution of dHJ, a very particular reaction that "dissolves the junctions by collapsing them into a catenated structure before releasing it by Topo3. In the field, dissolvasome and dissolution refers only to that function and authors should use "helices", "dismantling" and other terms when referring to the collective role of the three aforementioned helices.
2-The use of SDSA and DSBR to refer to sub pathways of the removal of DNA intermediates has to be use with caution. These concepts come from the two original models for HR, then sithetized to an integrated model. But one should incorporate too the processing of captured D-loops by nucleases and the fact that Srs2 may just prevent D-loop formation, not enforcing though a NCO outcome but rather a repair of the break by other means or using another template. These notions may have to be reinforced in the text.
3-The figures 2 and 3 reflect a very basic notion of windows of activity for the proteins discussed in the manuscript, but they don't reflect the gradient of activation states (i.e Mus81-Mms4 transition from active to hyper-active) and they don't reflect either the regulation of helicases or the more elusive motifs of regulation by sumo or ubiquitin. I will encourage the authors to integrate somehow this information in the figures with a more nuanced color-code and reflecting as well (maybe in a lateral column) if the actors are targeted or not by sumo and ubiquitin.
List of specific corrections and suggestions by line:
Line 18. Suggest to change "disengagement" to "removal". Lines 30 and 31. The sentence "Mirroring..." is too vague and needs re-writing. The sentence "Probably the most important..." is too speculative and in my opinion is not the role of HR to establish "physical interaction", it needs to be modified to reflect the "homology search and strand exchange capacity of HR able to read the information of homologous sequences to template repair and facilitate chromosome number reduction during meiosis". Line 36: change to "accurate recovery of an intact DNA molecule" Line 44: change "overhand" to "overhang" Line 45: the use of "endorse" seems not appropriate, consider changing to "are responsible of" or "catalyse" Line 48: the use of "sustain" seems not appropriate, consider just to remove the verb to "is required for full the full resection during ...." Line 54: change "launching" for "triggering" Line 59: Introduce "Dmc1 is a specialised player of HR in meiosis" instead. Do not abuse of "meiotic program" and use the simpler form "meiosis". Line 73: This section needs a little further precision: the nucleofilament has not only the ability to "invade" but importantly, to search and test homology by pairing sequences and then promote strand exchange. Line 77 and following paragraph. Instead of SDSA and DSBR as "strategies" it would be better to talk about "sub-pathways" Lines 79-80. I am unease with the use of "In turn, entails" for DSBR, it will be better to formulate as "Following the DSBR DSBR, the invading strand will be extended in the D-loop while the displaced strand will capture the other DSB end to ultimately form a closed covalent DNA intermediate known as the dHJ, that presents two physical connections between the two molecules of DNA involved" or something similar. Lines 81-82: Maybe it will need some modification. The two pathways may lead to different outcomes, but while SDSA leads exclusively to NCO, the DSBR sub-pathway can either generate crossovers and non-crossovers. Here, it will be probably wise to introduce the concept of early cleavage of the captured D-loop intermediate as a source of crossovers (as proposed early on by Matt Whitby and then also proposed and evidenced by the work of the Brill, Symington and Heyer lab among others) Line 88: Again, as in all the text, consider changing dissolution to "helicase-mediated displacement" and "nucleolytic resolution". Line 97: Title will be better reflecting the issue as "Processing of recombination intermediates by Helicases" In the Srs2 section, the concept of SDSA has to be curbed down to the sub-pathway leading to strand-displacement and strand-annealing. Thus, an early Srs2 role prevents SDSA to take place, and rather prevents HR all the way. Maybe a more nuanced discussion is necessary to highlight the possibility that Srs2 acts before synthesis occurs at the D-loop, and after synthesis. The interplay with Mus81-Mms4 will be placed outside SDSA too, then the importance of introducing the early cleavage of D-loop intermediates in the introduction. Line 166: The notion collaboration in the title of the section should be considered. The use of dissolution should be avoided. Line 170: change "execution" for "processing of" Line 171, genetic interactions are not between cells but between mutants or genes, remove "cells" Line 173, the use "crucial" is redundant and superfluous, re-write the sentence to state simply that Mph1 encodes for an ATP-dependent helices with a 3' to 5' polarity. Line 200: Rad52 foci resolution is not a good description considering the topic of the review. Use another term, not resolution. Maybe "dispersal"? Line 201: There is abundant reports of Mph1 meiotic roles in other organisms, maybe is better to tone down the first sentence or put it in context to yeast. In the paragraph of Sgs1 there are some orphan references listed by author's name not by number. Line 222, consider changing "employ" to "use" Line 241, replace "." after meiosis by "," and consider the us of "sustain" in that context. Line 248, change "obstruct" for "prevent" Line 254, avoid using expressions as "it is feasible to think" and "genome integrity pathway", there is no "genome integrity pathway" that we can recognise as by that name. Line 255, change the phrasing of "with a special relevance" to a neutral stance. Line 259, change "fist" to "first" Line 261: change to "via the two SIM motifs" to allow the possibility of the existence of other SIMs not identified. All phosphorylation and sumo regulations described for Sgs1 are not indicated in the figures 2 and 3. It may be interesting to add this to the figures in some way. Line 273: Consider changing to "Nucleolytic resolution" to tighten the scope. Line 277: The sentence starting by "This observation..." should be re-phrased. Line 281: change the sentence referring to Mms4 to be less categorical as it possess as well a critical role for the enzymatic function of the heterodimer, Mus81 being unable to cut the substrates by itself alone. Turn it the other way "most of the regulatory motifs of the heterodimer are found in Mms4". Line 289: The sentence "remaining recombination intermediates left undissolved" is not understandable, please re-phrase. Line 301: The notion of hyper-activation should be reflected in the figures 2 and 3. Line 348: change "unsolved" to "unknown". Line 387: It will be wise to introduce the notion suggested by Symington et al. in 2012/2013 of the presence of orphan HJ (single HJ) that can't be dissolved by Sgs1/STR and may become substrates for Yen1 in Anaphase when segregation of chromosomes reveals this remaining DNA links. Line 434: consider changing "binding platform" to the widely use term of "scaffold" Line 459: Consider discussing the fact that Rad1-Rad10 can also process D-loop intermediates and contribute to crossover formation in some DNA contexts (Symington's lab, Mazon et al 2012 NSMB) Line 529 and 544: avoid using repetitively "curiously", let the readers be curious by themselves. Line 547, please re-phrase avoiding using dissolution for all the helices-mediated reactions. Line 554, remove "headed" Line 556, "avoid using the term "dissolves complex" extensively beyond STR. Line 561: "depending on"
Author Response
We thank the reviewer for his/her constructive comments on our manuscript. We consider that the incorporation of these comments has significantly improved the quality of this review article.
Since the line numbering of the revised version has drastically changed with respect to the original manuscript, we consider that a point-by-point response referring the line number would be quite complicate for the reviewers. Please, refer to the “Track Changes” function in the Microsoft Word document containing the revised manuscript for the specific location of the modifications made.
Response to reviewer#1
Major conceptual points
1) As suggested by the reviewer, we have been used the terms “dissolvase/dissolution” only when referring to the STR complex, and “helicases/dismantling” for Mph1 and Srs2 throughout the text.
2) We have included both concepts suggested by the reviewer: The processing of captured D-loops by nucleases and the reversibility of D-loop formation. Appropriate references have been cited.
3) We have updated the figures with information regarding the modification of some enzymes by SUMO and ubiquitin. We have also modified the gradient of activation of Mus81-Mms4 in figures 2 and 3.
List of specific corrections and suggestions
We have addressed all specific corrections and suggestions raised by the reviewer. Please, see “Track changes” in the revised manuscript Word file for details.

Reviewer 2 Report
San-Segundo, Clemente-Blanco
Resolvases and disolvases in homologous recombination: …
The authors provide a well-written and comprehensive overview of the enzymes acting on joint DNA joint molecules during somatic and meiotic recombination. The manuscript is organized logically and provides, with some exceptions noted below, an excellent overview of the underlying literature. The figures are designed nicely and helpful to the reader, but some improvements could be made based on the comments below. Overall, this review will be a helpful summary of the literature, and the comments below are intended to strengthen an already excellent manuscript.
Main points:
1) Line 98, 166, 209, 273, 358, 434. 501: I really liked that the authors provided the historical background for the gene discoveries for SRS2, MPH1, and MLH1, because it gives the underpinning for their molecular function. However, the authors do not consistently apply this approach. I suggest providing the background for SGS1, MUS81, YEN1, and SLX1-4 as well, because it provides valuable information about their function. Line 209: SGS1 was discovered as a suppressor of slow growth of mutations in Top3, leading to the model that Sgs1 function produces the substrate for Top3. I think this fits the discussion by the authors well, and this original work should be cited (Gangloff 1994). Line 273: MUS81 was discovered in a two hybrid screen with RAD54. The authors discuss the functional interaction between both proteins and should also cite the original discovery (Interthal 2000). Line 358: GEN1/YEN1 was first identified in rice (Furukawa et al. 2003) and later in Drosophila (Ishikawa et al. 2004). The Drosophila study showed the broad substrate range of this nuclease, which is also the case for the yeast and human enzymes. This is a critical feature for the perceived late clean-up function of this nuclease. Line 434: The authors nicely discuss the discovery of Slx1-4 in the Sgs1 synthetic lethal screen by the Brill laboratory. However, the interpretation should consider their observation that this synthetic lethality is not suppressed by a recombination defect (Fricke and Brill 2003). This recombination-independent synthetic lethality would indicate that the cause of the synthetic lethality are not recombination intermediates.
2) Lines 108, 115-6, 135: The description of the Srs2 functions appears to be mixed up a little. Srs2 functions to strip Rad51 off ssDNA, an anti-recombinogenic function. Srs2 dissolves nascent D-loops, an anti-recombinogenic function. Srs2 dissolves D-loops extended by DNA polymerases, a pro-recombinogenic/pro-SDSA function. I do not think this becomes clear in the description in Line 108. In citing the activity of Srs2 to dissolve D-loops ref. 31 should be added to refs. 20-22 in line 108. Line 115: Ref. 28 reported the first purification of Rad55-57 and its function as a mediator in assembling Rad51 filaments; however, it did not report on counteracting Srs2. Line 115: The description of the Srs2 function in ref. 31 appears clouding the important point that Srs2 dissolves D-loops that are extended by Pol delta with preference when sumoylated PCNA is driving Pol delta compared to unmodified PNA. In discussing D-loop metabolism, also ref. 64 should be discussed here. This contribution was first to directly detect D-loops in somatic cells showing that loss of Srs2, STR (see below) and Mph1 (see below) function lead to accumulation of D-loops, providing the first in vivo evidence that these enzymes act on dissolving D-loops.
3) Line 140, Figure 2: The figure nicely summarizes data and concept regarding the enzymes discussed. However, I am not sure that the implications on the regulation are in all cases accurate, and I wished the authors would check some of their assertions about certain enzymes being inactive at a certain cell cycle time (maybe just less active). For example, the assertion that Mus81-Mms4 is inactive in S-phase in untenable. It is undisputed that Mus81-Mms4 is hyperactivated in late G2/M, but there is activity of the unphosphorylated form in vitro and the enzyme is active in vivo in S-phase as shown best by Mayle et al. The important contribution by Mayle et al. Science 2015 on this topic must be discussed and cited. The authors intrinsically acknowledge this on line 201, by characterizing the Cdc5, CDK, and DDK activation, as hyperactivation. I think the figure and the remainder of the discussion should be made consistent with this wording. I suggest deleting or rephrasing the sentence starting in line 307, because it is not broadly admitted that Mus81 is not active in S-phase. Another example is the MUS81-dependent cleavage of replication forks, which occurs after a long stall in wild type cells and fast in checkpoint-deficient cells (Hanada 2007, Kai 2005, Froget 2008).
4) Line 185: Ref. 64 showed that Mph1 counteracts accumulation of D-loops in vivo, which should be discussed and cited here, as it represents the first direct in vivo evidence of enzymes acting on D-loops.
5) Line 232: Ref. 64 showed that STR counteracts accumulation of D-loops in vivo, which should be discussed and cited here. It should be noted that in vitro (ref. 65) and in vivo (ref. 64) it is the Top3 decatenation activity that dissolves D-loops not the Sgs1 ATPase/helicase activity.
Additional points:
6) Line 40: Has it been shown that an individual DSB is processed 5’-3’ at both ends? Isn’t the current model for meiotic resection on endonucleolytic cleavage by the Mre11 complex and bi-directional resection 5’-3’ and 3’-5’? This may be too much detail, as the review is not focused on DSB resection.
7) Line 67, Figure 1: An important facet of recombination is the presence of metastable reversible intermediates, such as the RAD51 filament or the D-loop. Maybe this could be indicated in Figure 1 and referred to in the text, as the authors allude to reversibility of intermediates.
8) Lines 215, 220: Issues with referencing format.
9) Line 259: Change ‘fist’ to ‘first’.
10) Line 279, 434: The authors describe Mus81-Mms4 and Slx1-4 as structure-specific endonucleases, but discuss that both enzymes can cleave multiple DNA structures and are hardly specific. I suggest using the term structure-selective endonucleases which more accurately captures the substrate range of these enzymes.
11) Line 358, 417: In discussing the function of Yen1, the authors may want to carefully consider the difference between a physiological role in an otherwise wild type cell and a pathological role in mutant cells that present substrates in a context that is not physiological for the wild type. In the is regard the phrasing ‘backup’ (line 358) appears inaccurate. In order to be maintained during evolution, the YEN1 gene must contribute a selective advantage. Hence, there must a unique function of this enzyme besides functioning as a backup for other enzymes, unless the authors want to invoke that this function is required so often that becomes evolutionarily relevant.
12) Line 546-573: It seems that in the Concluding Remarks the authors are running out of steam. In particular the last sentences linking the topic to human disease and pathology are underdeveloped and the references given (152-154) are not very helpful and overly selective.
Author Response
We thank the reviewer for his/her constructive comments on our manuscript. We consider that the incorporation of these comments has significantly improved the quality of this review article.
Since the line numbering of the revised version has drastically changed with respect to the original manuscript, we consider that a point-by-point response referring the line number would be quite complicate for the reviewers. Please, refer to the “Track Changes” function in the Microsoft Word document containing the revised manuscript for the specific location of the modifications made.
Response to reviewer#2
Main points
1) We have included a brief description of the discovery of Sgs1, Mus81 and Yen1.
Previous comment on line 434: In contrast to the interpretation of the reviewer, we have found no evidence for a recombination-independent synthetic lethality between sgs1 and slx4 in the papers from the Brill laboratory (Fricke and Brill, Genes and Dev., 2003; Mullen et al., Genetics, 2001).
2) Following the suggestion of this reviewer, and also that of reviewer#1, we have added a sentence explaining the capacity of Srs2 for both eliminating nascent D-loops (an anti-recombinogenic activity) and for disrupting extended D-loops to promote SDSA (a pro-recombinogenic activity). The appropriate references, including old reference 31 (now reference 30), have been incorporated after this sentence.
We agree that previous reference 28 is not related to Srs2 regulation. We have cited this reference (now reference 39) in the appropriate place.
We have removed old reference 31 from the original sentence when discussing the regulation of Srs2 by PCNA-SUMO. Regarding previous reference 64, and also in agreement with the suggestion made by referee#1, we have included a paragraph in the introduction and modified figure 1 to include the contribution of Srs2, STR and Mph1 in the dismantling of D-loop structures. This reference is now cited in that paragraph (reference 13).
3) We have softened the assertion that Mus81-Mms4 is inactive in S-phase. We have also discussed possible situations where the complex may act during S-phase and added the references suggested by the reviewer regarding this issue.
4) We have inserted in the text the reference indicated by the reviewer (now reference 13).
5) We have specified that the processing of D-loop structures depends on Top3 decatenation activity. The previous reference 64 (now reference 13) is now cited in this context.
6) We have included the concept of bidirectional resection at both sides of the DSB and we have added the appropriate references in the text (Ref 1 and 2).
7) In agreement also with the suggestion raised by reviewer#1, we have discussed the concept of D-loop reversibility and modified figure 1 accordingly.
8) We have corrected the issue with the references.
9) We have corrected the mistake.
10) We have replaced the term “structure-specific endonucleases” with “structure-selective endonucleases”.
11) We have rephrased the sentences where the concept “backup” was used.
12) We have elaborated a little bit more on the discussion about the link between the DNA processing enzymes covered in this review and human diseases.

Round 2
Reviewer 2 Report
San-Segundo, Clemente-Blanco
Resolvases and disolvases in homologous recombination: …
Revised manuscript
Main Points
1) The descriptions of the discoveries of Sgs1, Mus81, and Yen1 are fine.
As for the recombination-independent lethality of the slx1 or slx4 sgs1 double mutants, please see below:
Verbatim citation from discussion page 1775 of Fricke & Brill 2003 Genes Dev.:
‘This model is in agreement with the following experimental results. First, it is consistent with the idea that Slx1–4 and Sgs1–Top3 interact upstream of homologous recombination. Previous studies indicated that Sgs1–Top3 and Mus81–Mms4 interact downstream of the initiation of recombination because sgs1 mus81 synthetic lethality is suppressed in strains lacking homologous recombination (i.e., sgs1 mus81 rad52 strains are viable; Fabre et al. 2002; Bastin-Shanower et al. 2003). In contrast, mutations in RAD52 failed to suppress the synthetic lethality of slx1 sgs1 or slx4 sgs1 strains, implying (1) that Sgs1–Top3 has more than one role in the cell, and (2) that homologous recombination, if required, is needed downstream of the Slx1–4/Sgs1–Top3 interaction.‘
Data for all slx genes are shown in Bastin-Shanower et al. 2003 in Figure 6 and cited as data not shown in Fabre et al. 2002.
The authors should discuss these data showing that the reason sgs1 slx1/slx4 are co-lethal is NOT dependent on HR and cite the relevant work. This important data show that Slx1/4-dependent processing of HR intermediates in sgs1 mutants in NOT the reason for the synthetic lethality. The data are consistent between the Brill laboratory (Bastin-Shanower et al. 2003) discussed in the Fricke and Brill 2003 paper and the Fabre/Heyer laboratories (cited as data not shown in Fabre et al. 2002)
2. The description of Srs2 is still incoherent. The Srs2 anti-rec activity comes first from stripping Rad51 from ssDNA and from reversing native D-loops. Srs2 anti-CO/pro-SDSA activity is based on dismantling extended D-loops. Hence, Srs2 acts on at least 3 levels not 2 as phrased on line 120. The authors should take greater care in describing the complexities of Srs2.
Author Response
Second revision
Response to reviewer#2
Main points
1) We apologize for having missed the idea of the recombination-independent lethality of slx1/slx4 sgs1 mutants. We have now incorporated this notion in the text and the appropriate references suggested by the reviewer.
2) We have mentioned now three levels of Srs2 action in the homologous recombination pathway.

Round 3
Reviewer 2 Report
The authors made the requested changes and additions.